# Utilization of Pollution Indices, Hyperspectral Reflectance Indices, and Data-Driven Multivariate Modelling to Assess the Bottom Sediment Quality of Lake Qaroun, Egypt

**Ali H. Saleh** [1], **Salah Elsayed** [2], **Mohamed Gad** [3], **Adel H. Elmetwalli** [4], **Osama Elsherbiny** [5], **Hend Hussein** [6], **Farahat S. Moghanm** [7], **Amjad S. Qazaq** [8], **Ebrahem M. Eid** [9,10,*], **Aziza S. El-Kholy** [10], **Mostafa A. Taher** [9,11] and **Magda M. Abou El-Safa** [1]

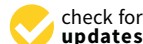



[1] Environmental Geology, Surveying of Natural Resources in Environmental Systems Department, Environmental Studies and Research Institute, University of Sadat City, Minufiya 32897, Egypt; ali.saleh@esri.usc.edu.eg (A.H.S.); magda.aboelsafa@esri.usc.edu.eg (M.M.A.E.-S.)

[2] Agricultural Engineering, Evaluation of Natural Resources Department, Environmental Studies and Research Institute, University of Sadat City, Minufiya 32897, Egypt; salah.emam@esri.usc.edu.eg

[3] Hydrogeology, Evaluation of Natural Resources Department, Environmental Studies and Research Institute, University of Sadat City, Minufiya 32897, Egypt; mohamed.gad@esri.usc.edu.eg

[4] Department of Agricultural Engineering, Faculty of Agriculture, Tanta University, Tanta 31527, Egypt; adel.elmetwali@agr.tanta.edu.eg

[5] Agricultural Engineering Department, Faculty of Agriculture, Mansoura University, Mansoura 35516, Egypt; osama_algazeery@mans.edu.eg

[6] Geology Department, Faculty of Science, Damanhour University, Damanhour 22511, Egypt; hendhussein@sci.dmu.edu.eg

[7] Soil and Water Department, Faculty of Agriculture, Kafrelsheikh University, Kafr El-Sheikh 33516, Egypt; farahat.ibrahim@agr.kfs.edu.eg

[8] Civil Engineering Department, College of Engineering, Prince Sattam Bin Abdulaziz University, Al Kharj 16273, Saudi Arabia; a.qazaq@psau.edu.sa

[9] Biology Department, College of Science, King Khalid University, Abha 61321, Saudi Arabia; mtaher@kku.edu.sa

[10] Botany Department, Faculty of Science, Kafrelsheikh University, Kafr El-Sheikh 33516, Egypt; aziza.elkhouli@sci.kfs.edu.eg

[11] Botany Department, Faculty of Science, Aswan University, Aswan 81528, Egypt

**\*** Correspondence: eeid@kku.edu.sa; Tel.: +966-54-574-1874

**Abstract:** Assessing the environmental hazard of potentially toxic elements in bottom sediments has always been based entirely on ground samples and laboratory tests. This approach is remarkably accurate, but it is slow, expensive, damaging, and spatially constrained, making it unsuitable for monitoring these parameters effectively. The main goal of the present study was to assess the quality of sediment samples collected from Lake Qaroun by using different groups of spectral reflectance indices (SRIs), integrating data-driven (Artificial Neural Networks; ANN) and multivariate analysis such as multiple linear regression (MLR) and partial least square regression (PLSR). Jetty cruises were carried out to collect sediment samples at 22 distinct sites over the entire Lake Qaroun, and subsequently 21 metals were analysed. Potential ecological risk index (RI), organic matter (OM), and pollution load index (PLI) of lake's bottom sediments were subjected to evaluation. The results demonstrated that PLI showed that roughly 59% of lake sediments are polluted (PLI > 1), especially samples of eastern and southern sides of the lake's central section, while 41% were unpolluted (PLI < 1), which composed samples of the western and western northern regions. The RI's findings were that all the examined sediments pose a very high ecological risk (RI > 600). It is obvious that the three band spectral indices are more efficient in quantifying different investigated parameters. The results showed the efficiency of the three tested models to predict OM, PLI, and RI, revealing that the ANN is the best model to predict these parameters. For instance, the determination coefficient values of the ANN model of calibration datasets for predicting OM, PLI, and RI were 0.999, 0.999, and 0.999, while they were 0.960, 0.897, and 0.853, respectively, for the validation dataset. The validation dataset

of the PLSR produced $R^2$ values higher than with MLR for predicting PLI and RI. Finally, the study's main conclusion is that combining ANN, PLSR, and MLR with proximal remote sensing could be a very effective tool for the detection of OM and pollution indices. Based on our findings, we suggest the created models are easy tools for forecasting these measured parameters.

**Keywords:** pollution load index; organic matter; potential ecological risk index; PLSR; MLR; ANN; SRIs; potentially toxic hazard

## 1. Introduction

The potential toxic metals (PTMs) assessment in the bottom sediment of lakes, particularly in the arid and semi-arid countries, has become a global topic with much attention from researchers [1,2]. The environmental pollution level has resulted from industry, agriculture, and unplanned urbanization activities determined through monitoring the degree of pollution in lakes [3,4]. Sediment monitoring, on the other hand, provides useful information on a variety of pollution indicators [5]. Sediments scrupulously store evidence of human activity and are crucial in determining pollution sources, history, dispersion, and damage to the ecosystem [6–8]. The lake's aquatic environmental conservation programs crucially need a clear explanation of the spatial distribution of PTMs in lake's bottom sediments and inferring the possible ecological risk [9,10]. The investigation of PTMs in sediments close to contaminated areas may perhaps be used to assess their effects on ecosystems and determine the hazards posed by waste pumped into the environment [2,11,12]. Potential toxic metals contaminants are usually absorbed and laid into sediments by particulate matter in the aquatic environment, and subsequently, the pollutants go back to the water bodies normally via the desorption mechanism [13,14]. Pollutants like PTMs in sediments have different distributions and concentration depending on particle size, silicates content, oxide-hydroxide content, carbonates, and organic matter content in sediments [15–17]. Saleh et al. [2] concluded that the main factors that may influence the distribution of metals in lakes' bottom sediment include: (1) organic matter percentages; (2) clay minerals; (3) salinity; (4) coagulations caused by the blending both fresh and salty water; (5) magnesium and iron oxides; (6) processes of oxidation and reduction; and (7) metal sources.

Lake Qaroun is one of the Egyptian Western Desert's most prominent geomorphological features. Apart from serving as a natural discharge location for El-Fayoum district, it is a destination for migratory birds, fisheries, salt production, and tourism activities, in both winter and fall seasons [2,11,18]. In 1989, the Prime Ministerial Decree No. 943/1989 [19] established Lake Qaroun as a natural protectorate, in accordance with Law 102/1983. Lake Qaroun annually received a vast volume (<0.45 billion cubic meter per year) of untreated effluents from agricultural, domestic, industrial, and aquacultural activities [20–22]. The lake's water is only lost through evaporation processes, causing a steady increase in its water salinity and pollution content [21,23]. So, the lake's environmental quality is sensitive and threatened by the accumulation of pollutants, affecting the food chain and ecosystem.

The quality and sustainability of aquatic ecosystems have been evaluated by several geochemical, statistical, and artificial methods, which assess the environmental risk of PTMs in bottom sediments [2,24–26]. The potential ecological risk index (RI) and pollution load index (PLI) are two of the most important indicators for assessing the environmental combined danger of several PTMs in bottom sediments [27]. Environmental data is often processed using geochemical pollution indices and multivariate modelling to identify probable pollution sources that impact different aquatic systems. It is a well-known method of natural-resource management that aids in the selection of the optimum pollution remedies [28]. The combination of these approaches can significantly increase the accuracy of PTMs contamination assessments in bottom sediments [29]. As technology progresses and expands its functions, newly developed approaches for monitoring aquatic ecosystems'

quality are employed. Cutting-edge technologies, such as spectral analysis of bottom sediments and artificial modelling, utilize aquatic ecosystems evaluation and monitoring to minimize money and time while enhancing accuracy [26]. Consequently, the Lake Qaroun ecosystem quality should be regularly checked to take appropriate precautions against any environmental hazards, evaluate current protection and remediation projects at the lake, and support decision-makers in protecting this nature reserve from potential ecological risks.

Grain size and the amount of organic matter (OM) are two of the most important parameters in determining the distribution of PTMs in sediments [15,30]. Higher PTM concentrations are seen in fine-grained sediments [31], which are linked to higher magnetic susceptibility. Organic matter, according to Farhat and Aly [32], was more significant in governing PTM distribution. Organic matter was also found to be a significant indicator for ecological risk [33]. Saleem et al. [34] revealed that OM may preserve PTMs in sediments and play a crucial role in bottom sediment quality evaluation. Organic matter is important in several physical and chemical processes in the sediment environment, and thus it has a significant influence on sediment spectral reflectance characteristics.

Sediment OM, PLI, and RI estimation has always been based entirely on point-sampling and laboratory tests. Although this approach is slightly accurate, it is not fast, expensive, damaging, and spatially constrained, making it ineffective for monitoring these parameters. Remote sensing approaches have recently been used to estimate OM, since visible (VNIR) spectroscopy is a non-destructive, quick, and repeatable tool for quantifying soil's chemical, physical, and biological properties [35,36]. Vibrations in the bonds between various atoms (C, N, H, O, P, and S) produce reflectance signals. Weak overtones and inter-actions of basic vibrations dominate the NIR range (700–2500 nm) and electronic transitions (400–700 nm) regions of the electromagnetic (EM) spectrum as a result of the stretching and bending of OH, NH, and CH groups [37]. Many researchers (e.g., Chang and Laird [38]; Bartholomeus et al. [39]; Barthès et al. [40]; Viscarra Rossel et al. [41]; Cécillon et al. [42]; Franceschini et al. [43]; Vaudour et al. [44]) have noted the advantages of the VIS and mid-infrared spectral range in terms of a shorter cycle and less cost for quantifying OM in comparison to traditional OM estimation procedures, which may lead to significant cost savings [45–48].

The correlation coefficient curve between OM contents and spectral reflectance can be used to find the sensitive spectral area. As a result, OM spatial estimate can be achieved by creating a link between real OM contents for every sample and fundamental band information of selected images (e.g., digital number (DN), reflectivity, power func-tions/derivatives/logarithms based on original reflectivity) [49–51]. OM dominates the VIS range (400–700 nm) and NIR (700–2500 nm) wavelength ranges [52,53]. Recently, relative studies have frequently incorporated a variety of environmental parameters (e.g., terrain, humidity, and vegetation) in conjunction with the fundamental universal bands mentioned above, with the goal of improving the spatial accuracy of OM mapping [54,55]. The ra-tio vegetation index (RVI), normalised difference vegetation index (NDVI), soil-adjusted vegetation index (SAVI), and greenness vegetation index (GVI), had greater efficiency than the many indices to assess OM content [51,56]. For example, the NDVI extracted from the Sentinel-2A satellite's reflectance simulated the OM in India's Sariska Tiger Re-serve; and for single-phase synchronisation data, the $R^2$ reached 0.74 [57]. OM was found to be linked with reflectance range of 500–1200 nm according to Mathews et al. [58], al-though Beck et al. [59] claimed that the 900–1220 nm zone is suitable for mapping OM. Krishnan et al. [60] predicted OM content using a slope parameter of roughly 800 nm.

Although there are many previous studies available to estimate OM using spectral reflectance indices (SRIs), there is very little information about estimating pollution indices such as PLI and RI using SRIs. Variable deterministic models have been used in this area for decades [61]. However, the statistical performance of varying deterministic models is typically low because in reality natural ecosystems are typically too complicated for state-of-the-art models.

ANN, PLSR, and MLR might be used to create models for estimating sediment quality indicators in lake systems quickly and accurately. Recently the ANNs demonstrated high performance as a regression method when used for pattern recognition and function determination [62]. In comparison to traditional methods, an ANN can tolerate and interpret an incomplete dataset; approximate results as well as are less vulnerable to outliers [63]. As a result of its tremendously parallel processing architecture, ANNs can effectively handle complicated calculations and therefore are among the most preferred techniques for high-speed processing of massive datasets [64]. As a machine learning approach, ANN has increasingly been used to address environmental issues [65]. Technologies such as remote sensing, Internet of Things (IoT), Unmanned Aerial Vehicles (UAVs), Big Data Analytics (BDA), and Machine Learning (ML) are especially promising and have the potential to revolutionize environmental monitoring methods [66]. IoT is one of the most innovative technologies in modern wireless communications [67,68]. The fundamental principle is the interaction of various physical entities or devices linked to the Internet via specialised addressing systems. IoT technology may be used in a variety of industries, including manufacturing, transportation, healthcare, cars, smart homes, agriculture, and environment [69,70].

In addition, the methods of expressing a linear connection between different variables (independent and dependent) are known as PLSR and MLR [71,72]. These models are able to aggregate data from SRIs into a single index to increase the predictability of an estimated variable. One of the most important features of PLSR is its ability to reduce many collinear components to a few non-correlated latent parameters, reduce duplicate data, and also limit overfitting and underfitting to a minimum [73,74]. Based on the advantages of ANN, PLSR, and MLR, the OM and two polluted indices (PLI and RI) may be estimated concurrently from many SRIs using these approaches.

Very little attention has been given to assess the efficacy of ANNs, PLSR, and MLR models combined with SRIs for quantifying OM, PLI, and RI. Therefore, the overall goal of this research was to provide useful tools for making informed decisions regarding the sediment quality of Lake Qaroun. The premise behind this research was that different SRIs groups' machine learning modelling (ANNs) and multivariate models (PLSR and MLR) may be useful tools for measuring OM, PLI, and RI of sediment in Lake Qaroun.

Consequently, the particular goals of this study were to: (i) determine the OM, pollution rate, and environmental hazards of possibly toxic elements (PTMs) in the bottom sediment of Lake Qaroun via the calculation of PLI and RI; (ii) extract the optimized NSRIs-2b and NSRIs-3b for OM, PLI, and RI using the 2-D and 3-D-dimensional slice map; (iii) assess the accuracy of three distinct SRI groups in measuring OM, PLI, and RI of bottom sediments; and (iv) based on selected SRIs, estimate the accuracy of ANNs, MLR, and PLSR models in quantifying the OM, PLI, and RI of bottom sediments.

## 2. Materials and Methods

### 2.1. Study Area

Lake Qaroun is an oval natural inland saline lake with an average length of 45 km east to west and a width of 5.7 km south to north, with a water depth of 1–8.8 m. It is situated in the El-Fayoum Depression, Western Desert, some 95 km south-west Cairo, Egypt. It exists between $30°24'$ and $30°50'$ E longitudes and $29°24'$ and $29°33'$ N latitudes (Figure 1). The water body of the lake occupies a surface area of 235 km$^2$. Several land use and human activities (e.g., agricultural, industrial, and entertainment) surround the lake on the east and south. In addition, there are urbanized regions and a well-developed transportation network. The two drains (El-Bats and El-Wadi), which together discharge 338 million m$^3$/year, provide the majority of the water in Lake Qaroun [22]. Further, there are 12 secondary drains at the lake's southern side, which also discharge water into the lake (Figure 1).

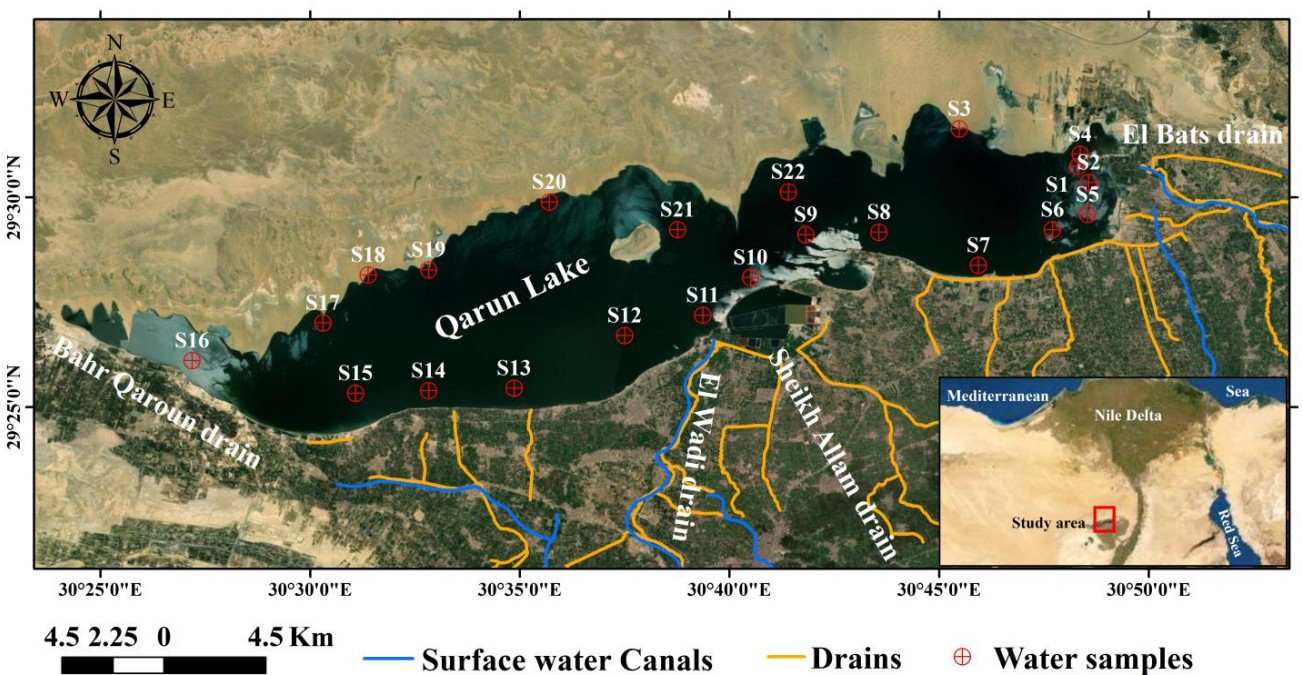

**Figure 1.** Location map of Lake Qaroun and different sampling points throughout the entire lake.

El-Fayoum Depression digs include gyps-ferrous shale, limestone, white marls, and sandstone from Middle Eocene rocks [75]. Lake Qaroun is encompassed with eolian, nilotic (alluvial sediments), and lacustrine layers of Quaternary deposits. In alluvial deposits, sands and gravels in different sizes mix with calcareous clay and silt content. The lacustrine deposits compose ferruginous sandy silt intercalated with claystone, gypsum, and calcareous minerals [76]. Fine particles (>63 microns) made up nearly 70% of the bottom sediments on the eastern and southern sides of Lake Qaroun. The lake water salinity ranged from 28 to 40 g/L, the surface temperature ranged from 22 to 25 °C, and the while drain water salinity ranged from 0.6 to 2 g/L. Heavy minerals made up more than 1% of the sand dunes [21,77,78].

Qaroun Lake's climate is characterized by a hot summer and cold winter. So in winter, the temperature and surface wind speed reach their minimum degree, and the humidity and the rainfall reach their maximum. While in summer, the maximum temperature records and the minimum degree of moisture and rainfall are noted. Generally, the water level and the water salinity in the lake increase in winter and decrease in summer.

### 2.2. Sampling and Analyses

A Van Veen Grab sampler was employed to collect representative bottom sediment samples [79] in autumn 2018 and 2019 from depths between 2 and 5 m at 22 different locations to cover the entire lake (Figure 1). Three sediment samples were gathered at each site. The sampling location's co-ordinates were identified with a hand-held GPS (Garmin/eTrex Vista HCx/personal navigator). Once the samples were collected, they were preserved in tight plastic bags and then taken to the laboratory in an ice-box for analysis. The sediments were left to dry out at room temperature to achieve a constant weight prior to sieving through 2.0 mm sieves. It was then ground using an agate mill (Retsch RM200) before being kept in a little glass bottle for analysis.

The total organic matter content (TOM) in sediment samples was determined using the loss-on-ignition (LOI) procedure as described by Allen et al. [80]. Each sediment sample was digested using a Speedwave microwave digestion device, according to the US EPA 3052 procedure [81]. The digested solution was filtered and diluted in a measuring flask to 50 mL with deionized water. Quality control procedures utilized included replicating samples and using standard sediment reference substances (GBW07333) given by China's

Second Institute of Oceanography (SOA). The used glass was pre-cleaned and immersed in dilute nitric acid for a minimum of 24 h, followed by soaking and rinsing with deionized water to help avoid test vessel contamination. A blank was prepared in the same method and with the same reagents for precision testing. Inductively coupled plasma mass spectra (ICP-MS) (ICAP TQ ICP-MS Thermo Fisher Scientific Inc., Waltham, MA, USA) was applied to define the concentrations of 21 PTMs. The 21 PTMs of each sample were analysed as indicated in Tables S1 and S2 over 2 years. The laboratory procedures were run at the University of Sadat City's (Environmental and Food Lab (EFL); Environmental Geology Lab (EGL)- ISO/IEC 17025/2017).

### 2.3. Environmental Pollution Indices

The Lake Qaroun bottom sediment was evaluated by applying ecological risk assessment methodologies such as PLI and RI depending on the PTMs concentration in sediments.

#### 2.3.1. Pollution Load Index

The PLI is one of the successful methods to determine the severity of sediment pollution. The Equation (1) proposed by Tomlinson et al. [82] was used to calculate PLI for each location.

$$\text{PLI} = (\text{Cf}_1 \times \text{Cf}_2 \times \text{Cf}_3 \times \ldots \times \text{Cf}_n)^{1/n} \tag{1}$$

where Cf refers to the contamination factor for each tested metal in sediment; n refers to the number of PTMs analysed in every sample ($n = 21$). Based on the PLI results, the tested samples can be split into two main groups: non-polluted (PLI < 1) and polluted (PLI > 1), as presented in Table 1.

#### 2.3.2. Potential Ecological Risk Index

The RI explores the potential ecological threat caused by PTMs contained in sediment by measuring the sensitivity of varying biological populations to harmful metals [83,84]. Håkanson's [85] Equations (2) and (3) were used in the RI calculation.

$$\text{RI} = \sum_{1}^{n} \text{Er} \tag{2}$$

$$\text{Er} = \text{Tr} \times \text{Cf} \tag{3}$$

where Er is the individual element's potential RI, and Tr is the metals' toxic reaction factor proposed by Håkanson [80]. The PLI is divided into four categories, ranging from minimal to very high ecological risk as summarized in Table 1.

**Table 1.** Classes of pollution load index (PLI) and potential ecological risk index (RI).

| Contamination Indices | Values | Classes | Reference |
|---|---|---|---|
| PLI | 1 > PLI<br>1 < PLI | Unpolluted<br>Polluted | [86] |
| RI | RI < 150<br>150 < RI < 300<br>300 < RI < 600<br>600 < RI | Low ecological risk<br>Moderate ecological risk<br>Considerable ecological risk<br>Very high ecological risk | [85] |

### 2.4. In Situ Ground-Based Reflectance Measurements

The reflected spectra of each sediment sample were acquired using a mobile spectrometer (tec5 AG, Oberursel, Germany) at VIS and NIR wavelengths with a spectral range of 302–1148 nm, which was interpolated to 2 nm final spectral resolution, yielding a total of 424 data values per spectrum curve. The device is made up of two basic units; the first unit was linked to a diffuser, which was responsible for measuring radiation as a reference signal.

Sediment samples were kept in 10 cm diameter black Petri dishes. The spectrometer had a scanning area of 0.05 m$^2$ when held at 25 cm above sample and in a vertical position (nadir position). To the fluctuations in sun zenith angle at a minimum, spectral measurements were obtained around midday. Spectral measurements were duplicated three times at each surface sediment sample, for a total of 15 scans. The mean value of three measurements was calculated to derive the measured spectrum for a single surface sediment sample. To calibrate the spectra collected by the spectrometer, a white reference standard (Apolyte-trafluoroethylene white Spectralon reflectance panel) was utilized. Finally, noise at the two ends of the spectrum was removed by smoothing the spectral reflectance.

*2.5. Selection of Newly Constructed and Previously Published Spectral Reflectance Indices*

Table 2 shows the formula and references for various SRIs. Eight published spectral indices (PSRIs) and 24 newly extracted SRIs were assessed in this research as presented in Table 2. Two and three band ratios (NSRIs-2b and NSRIs-3b) were derived using 2-D and 3-D correlogram maps. Different 2-D and 3-D correlogram maps were created using MATLAB 7.0 (The MathWorks, Inc., Natick, MA, USA). 2-D and 3-D correlogram maps aimed to present the determination coefficient ($R^2$) values of the linear regression between measured parameters and potential interactions between any two wavelengths in the entire spectral range for NSRIs-2b (Figure 2) and possible interactions between any three wavelengths in the visible region (VIS) and red-edge (Red) region from 390 to 750 nm for NSRIs-3b (Figure 3).

As indicated in Table 2, a ratio spectral index the NSRIs-2b was calculated according Elsayed et al. [26]:

$$RSI = R_1/R_2 \tag{4}$$

$R_1$ and $R_2$ refer to the values of spectral reflectance at various wavelengths.
As indicated in Table 2, the NSRIs-3b was calculated as a normalised difference index:

$$NDI = (R_1 - R_2 - R_3)/(R_1 + R_2 + R_2) \tag{5}$$

$R_1$, $R_2$, and $R_3$ represent the values of spectral reflectance at selected wavelengths

**Table 2.** Descriptions commonly used in the literature and newly derived spectral indices examined in this study.

| Spectral Reflectance Indices | Formula | References |
|---|---|---|
| PSRIs | | |
| Average Blue (450 to 486 nm) | $\rho_{Blue}$ | [87] |
| Average Red (620 to 750 nm) | $\rho_{Red}$ | [87] |
| Average NIR average (700 to 1100 nm) | $\rho_{NIR}$ | [87] |
| Normalized difference vegetation index (NDVI$_{780-550}$) | $(R_{780} - R_{550})/(R_{780} - R_{550})$ | [88] |
| Red-Normalized difference vegetation index (R-NDVI) | $(\rho_{NIR} - \rho_{Red})/(\rho_{NIR} + \rho_{Red})$ | [89] |
| ratio vegetation index (RVI) | $\rho_{NIR}/\rho_{Red}$ | [87] |
| Difference vegetation index (DVI) | $\rho_{NIR} - \rho_{Red}$ | [90] |
| Optimized soil-adjusted vegetation index (OSAVI) | $(1 + 0.16) \times (\rho_{NIR} - \rho_{Red})/\rho_{NIR} + \rho_{Red} + 0.16$ | [91] |
| NSRIs-2b | | |
| Ratio spectral index | | |
| RSI$_{674,666}$ | $R_{674}/R_{666}$ | Present study |
| RSI$_{660,680}$ | $R_{660}/R_{680}$ | |
| RSI$_{676,664}$ | $R_{676}/R_{664}$ | |
| RSI$_{800,650}$ | $R_{800}/R_{650}$ | |
| RSI$_{750,740}$ | $R_{750}/R_{740}$ | |
| RSI$_{680,664}$ | $R_{680}/R_{664}$ | |
| RSI$_{1080,650}$ | $R_{1080}/R_{650}$ | |

**Table 2.** *Cont.*

| Spectral Reflectance Indices | Formula | References |
|---|---|---|
| $RSI_{822,642}$ | $R_{822}/R_{642}$ | |
| $RSI_{1078,656}$ | $R_{1078}/R_{656}$ | |
| $RSI_{666,670}$ | $R_{666}/R_{670}$ | |
| $RSI_{900,1042}$ | $R_{900}/R_{1042}$ | |
| $RSI_{912,1028}$ | $R_{912}/R_{1028}$ | |
| NSRIs-3b | | |
| Normalized difference spectral index | | |
| $NDSI_{682,666,692}$ | $(R_{682} - R_{666} - R_{692})/(R_{682} + R_{666} + R_{692})$ | Present study |
| $NDSI_{682,692,666}$ | $(R_{682} - R_{692} - R_{666})/(R_{682} + R_{692} + R_{666})$ | |
| $NDSI_{684,666,692}$ | $(R_{684} - R_{666} - R_{692})/(R_{684} + R_{666} + R_{692})$ | |
| $NDSI_{684,692,662}$ | $(R_{684} - R_{692} - R_{662})/(R_{684} + R_{692} + R_{662})$ | |
| $NDSI_{656,664,676}$ | $(R_{656} - R_{664} - R_{676})/(R_{656} + R_{664} + R_{676})$ | |
| $NDSI_{682,692,664}$ | $(R_{682} - R_{692} - R_{664})/(R_{682} + R_{692} + R_{664})$ | |
| $NDSI_{682,690,666}$ | $(R_{682} - R_{690} - R_{666})/(R_{682} + R_{690} + R_{666})$ | |
| $NDSI_{682,666,690}$ | $(R_{682} - R_{666} - R_{690})/(R_{682} + R_{666} + R_{690})$ | |
| $NDSI_{682,668,692}$ | $(R_{682} - R_{668} - R_{692})/(R_{682} + R_{668} + R_{692})$ | |
| $NDSI_{680,688,672}$ | $(R_{680} - R_{688} - R_{672})/(R_{680} + R_{688} + R_{672})$ | |
| $NDSI_{680,666,688}$ | $(R_{680} - R_{666} - R_{688})/(R_{688} + R_{666} + R_{688})$ | |
| $NDSI_{658,650,670}$ | $(R_{658} - R_{650} - R_{670})/(R_{650} + R_{666} + R_{670})$ | |

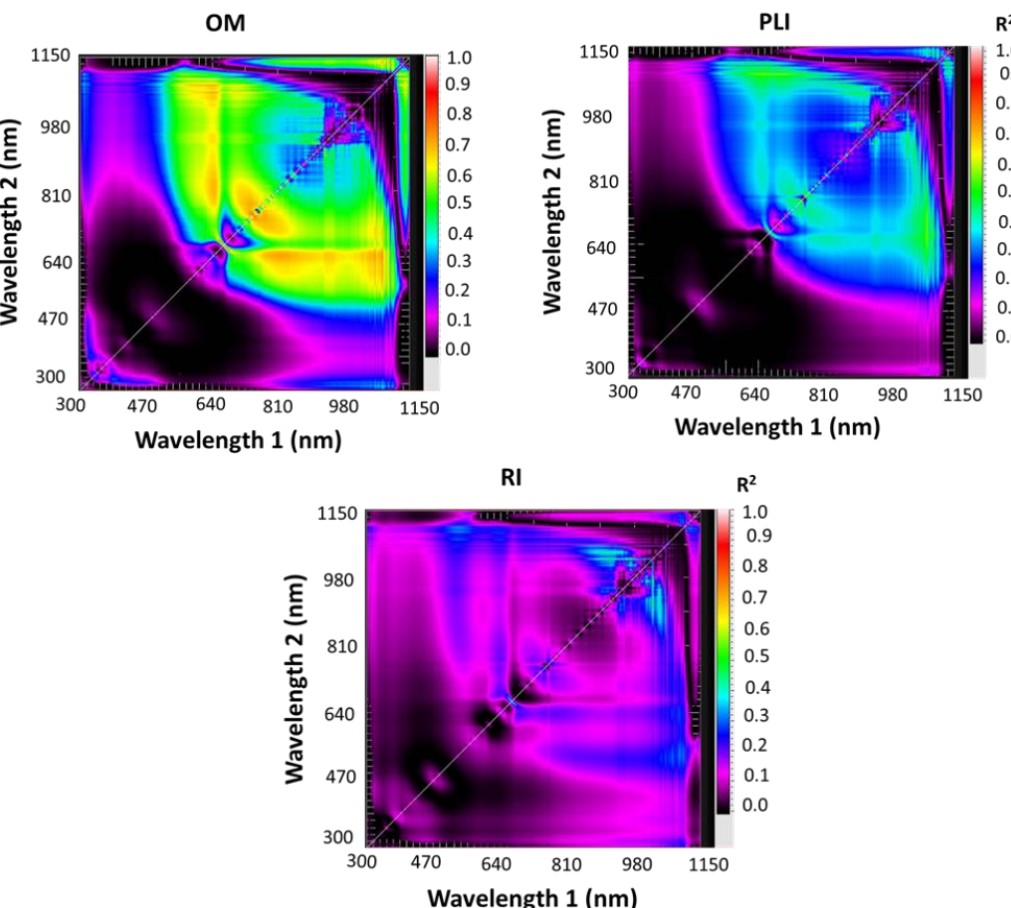

**Figure 2.** Correlation matrices showing the ($R^2$) values for prospective dull wavelength combinations of the spectra range from 302 to 1148 nm with organic matter (OM), potential ecological risk index (R1), and pollution load index (PLI) across 2 years.

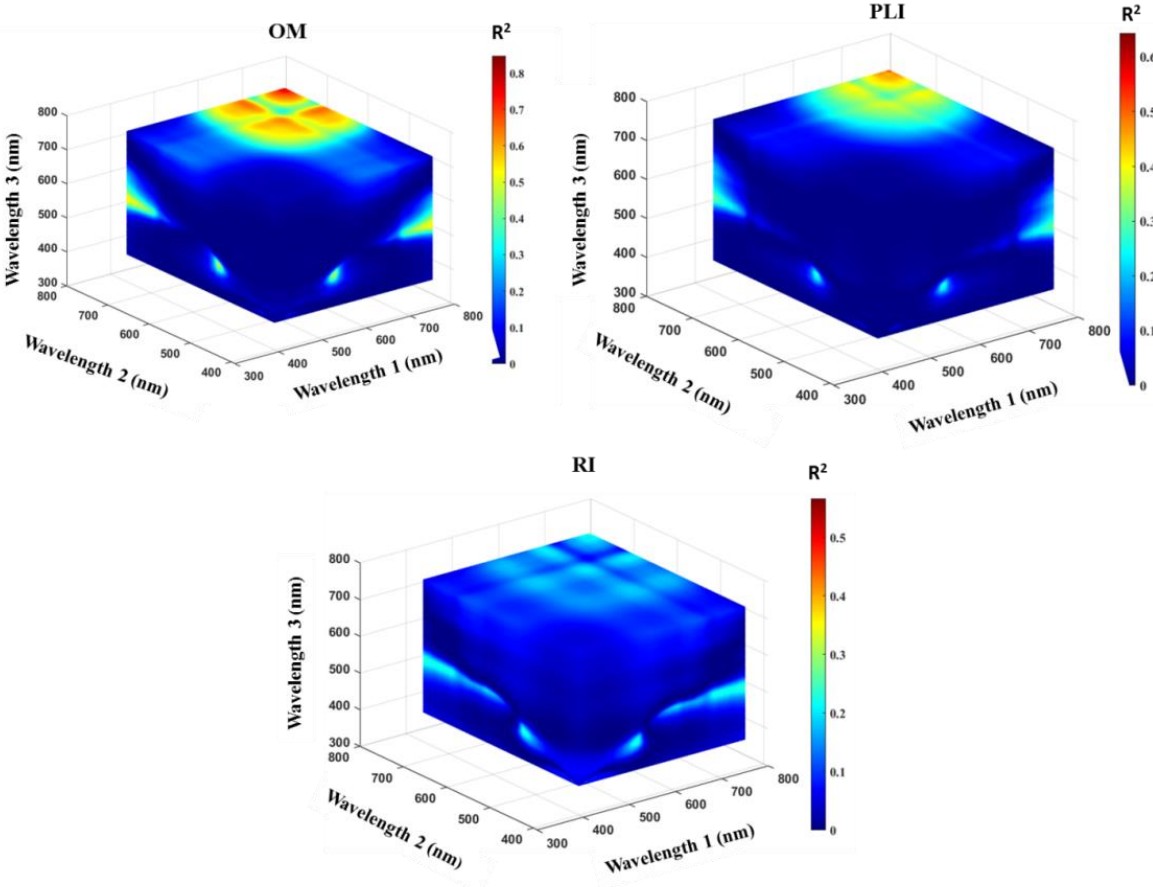

**Figure 3.** Three-dimensional slice map, calculated for all potential three-band combinations from 390–750 nm, of the $R^2$ values obtained for the association between organic matter (OM), potential ecological risk index (R1), and pollution load index (PLI) across 2 years.

*2.6. Back-Propagation Neural Network (BPNN)*

The BPNN is one of the most widely utilized artificial neural network models in the backpropagation neural network [92]. There are three types of layers in the BPNN: (1) the input layer is primitive data for the neural network, (2) the concealed layer is an intermediate layer between independent input and dependent output layers, and (3) the output layer offers the results of the identified inputs. The ANN is a machine-learning technique that uses multiple layers to extract elevated-level features from low-level input. The network has two hidden layers, with the number of nodes determined by the efficiency of the regression. The "activation" nodes are represented by the hidden layers, which are commonly labeled as weight. The final layer represents the output layer, which displays the measured parameter's anticipated value. ANN models are extrapolated mathematical-based models that employ a sequence of nodes or neurons connected by rated connections to replicate human cognition in pattern recognition and prediction [93,94].

At least 2000 iterations were used in training the network, or until the error reached a value less than $10^{-4}$. The validation procedure with the LOOV method was used on the training data set to identify the number of neurons in the concealed layer for this model. The parameter of restricted memory (lbfgs) was used as a weight optimizer to perform the algorithm more effectively [95]. The following formula was used to find the most relevant

feature to increase the regression model's predictive capacity and reduce hyperspectral image dimensionality [96]:

$$M = \frac{\sum_{j=1}^{n_H}\left[\left(|I|_{P_j} / \sum_{k=1}^{n_p}|I|_{P_{j,k}}\right)|O|_j\right]}{\sum_{i=1}^{n_p}\left(\sum_{j=1}^{n_H}\left[\left(|I|_{P_{i,j}} / \sum_{k=1}^{n_p}|I|_{P_{i,j,k}}\right)|O|_j\right]\right)} \tag{6}$$

where $M$ is the important estimate of the input variable, $n_p$ represents the number of input variables, $|I|_{P_j}$ is the absolute value of the unseen layer rate matching to the $p$th input variable and the $j$th hidden layer, $n_H$ refers to the number of unseen layer nodes, and $|O|_j$ is the absolute value of the output layer rate consistent with the $j$th unseen layer.

*2.7. Multiple Linear Regression (MLR) and Partial Least-Square Regression (PLSR)*

In this work, both PLSR and MLR statistical models were evaluated as innovative techniques to assess the OM two pollution indices (PLI, and RI). Both models were built with the unscramble X software in V. 10.2. (CAMO-Software AS, Oslo, Norway). Both models included the best SRIs, which were selected by ANN model as input variables (independent variables) for the prediction of OM, PLI, and RI as output variables (dependent variables). To link between the inputs and outputs, PLSR was run in tandem with the LOOCV. One of the critical stages in PLSR analysis is to identify the right number of latent variables (LVs) to show the calibration data while avoiding both overfitting and underfitting. The procedure of random ten-fold cross-validation was performed on the datasets to enhance the reliability of the obtained results. The least-square technique was run to determine the parameters for the MLR, which minimised the total of errors squared.

*2.8. Model Evaluation*

The following statistical metrics were used to evaluate the performance of a regression model: coefficient of determination ($R^2$) and root mean square error (RMSE) [97,98]. The following are the explanations for all parameters: $F_{ave}$ is the average value, $F_{act}$ is the actual value that was quantified from laboratory calculations, $F_p$ is the forecast or simulated value, and $N$ is the total number of data points.

Root mean square error

$$\text{RMSE} = \sqrt{\frac{1}{N}\sum_{i=1}^{N}\left(F_{act} - F_p\right)^2} \tag{7}$$

Coefficient of determination

$$R^2 = \frac{\sum\left(F_{act} - F_p\right)^2}{\sum\left(F_{act} - F_{ave}\right)^2} \tag{8}$$

## 3. Results and Discussion

*3.1. Metals Distribution*

The average metal concentrations in sediment samples gathered from Lake Qaroun through years of investigation, 2018 and 2019, are shown in Tables S1 and S2. The sediments on the lake's eastern side have the greatest concentrations of metals. The western side sediments reported the lowest metals concentration values. The eastern and centre regions of the lake (near the exits of El-Wadi and El-Bates drains) contain higher contents of OM than the western side. PTMs distributions in sediments vary depending on particle size, oxide-hydroxide content, and OM content [15,17]. Previously, El-Zeiny et al. [22], El-Kady et al. [11], and Saleh et al. [2] emphasized a similar observation. All PTMs in the lake sediment samples appeared to be accumulated at the exits of El-Bats and El-Wadi drains, indicating high concentrations in the eastern and central parts of the lake. Abdel Wahed et al. [20] demonstrated a link between PTMs found in lake sediments and those found in drain water. As a result, El-Bats outflow is the lake's primary source

of environmentally sensitive metals, particularly in the lake's eastern and north-eastern sections. El-Sayed et al. [9]; Attia et al. [18]; El-Kady et al. [11]; and Saleh et al. [2] previously established the susceptibility of the southern and eastern sides of the lake due to metal pile up in lake sediments. Coagulation at the mouth of drains that flow into the lake, caused by the mixing of freshwater and saltwater, will result in significant sedimentation at the drains' mouths, increasing the percentage of fine particles and OM [99]. A diminishing aquatic ecology can be found in areas with high content of organic materials, like the areas under examination near drain inlets. Reducing zones may have low dissolved metals as a result of microbial reduction, allowing PTMs to precipitate [100].

*3.2. Assessment of Organic Matter, Pollution Load Index (PLI) and Risk Index (RI) of Bottom Sediment in Lake Qaroun*

Over 2018 and 2019, shipboard field visits were conducted to collect sediment samples to identify the OM, PLI, and RI. Table 3 shows the descriptive statistical analysis of the OM and two environment pollution indices of sediment samples. According to the obtained results, the three measured indicators are significantly different from one site to another. The content of OM is obviously higher on the lake's eastern side than on the lake's western side. As presented in the table, the content of OM differed from one site to another with a minimum content of 0.72% at site 19 on the western side of the lake and a maximum value of 17.97% at site 9 on the eastern side. At sites 1 through 5 the OM was higher than those from sites 10 through 22. These increases in the OM content in the lake's eastern side may be attributed to the elements that come through El-Bats drain. Like the results obtained for the OM, PLI had the same trend since the greatest value of PLI was recorded at site 9 while the minimum record of PLI was obtained at site 17. Again, the PLI was recorded higher on the eastern side of the lake in comparison to the western side with a few exceptions (e.g., at site 21). Regarding RI, its values significantly varied from one point to another with respective minimum and maximum values of 808.24 and 1515.8 obtained at sites 16 and 9. As plotted in Figure 4, PLI data revealed that around 59% of Lake Qaroun is polluted (PLI greater than 1), especially samples of southern and eastern sides of the samples at the central location of the lake, while the 41% that is unpolluted (PLI < 1) is composed of samples of the western and western northern regions, and thus actions must be taken to minimize the harmful effects of pollution since the lake is among the main sources to several living organisms. According to the RI's findings, all the examined sediments pose a very high ecological risk (RI > 600) (Figure 4). Human enterprises including entertainment, industrial, agricultural, aquaculture, and urban, all of which surround Lake Qaroun, are thought to be possible sources for hazardous metal enrichment in lake sediments. Continuing to dispose these wastes without proper treatments leads to elevated proportions of harmful metals in the lake sediments, which in the end causes the deterioration of the lake's various ecological systems, a full end of fishing and other recreational activities, and also a direct effect on migratory birds.

**Table 3.** Statistical description of some sediment quality parameters in Lake Qaroun over the 2-year study.

| | OM (%) | SD (%) | PLI | SD | RI | SD |
|---|---|---|---|---|---|---|
| Site 1 | 14.38c | 0.212 | 1.27bc | 0.008 | 1301.4bc | 58.419 |
| Site 2 | 16.31b | 0.028 | 1.24b–d | 0.032 | 1240.3cd | 24.316 |
| Site 3 | 12.43d | 0.141 | 1.11b–g | 0.128 | 1058.6f–h | 32.500 |
| Site 4 | 12.47d | 0.320 | 1.24b–d | 0.057 | 1106.5e–g | 89.238 |
| Site 5 | 11.90e | 0.151 | 1.27bc | 0.059 | 1220.6c–e | 35.169 |
| Site 6 | 4.24j | 0.184 | 1.15b–f | 0.031 | 1128.1d–f | 76.646 |
| Site 7 | 3.08lm | 0.212 | 0.91g–h | 0.076 | 932.45h | 87.842 |
| Site 8 | 3.11lm | 0.086 | 0.86h–i | 0.301 | 1289.7bc | 3.791 |
| Site 9 | 17.97a | 0.147 | 1.65a | 0.045 | 1515.8a | 39.498 |
| Site 10 | 5.80i | 0.071 | 1.07b–h | 0.044 | 992.3h | 4.913 |

**Table 3.** *Cont.*

|  | OM (%) | SD (%) | PLI | SD | RI | SD |
|---|---|---|---|---|---|---|
| Site 11 | 8.33h | 0.283 | 1.039c–h | 0.092 | 1122.3d–g | 33.086 |
| Site 12 | 2.66mn | 0.071 | 1.02d–h | 0.197 | 1065.1fg | 17.422 |
| Site 13 | 9.59g | 0.530 | 1.20b–e | 0.041 | 1386.1b | 66.197 |
| Site 14 | 3.41kl | 0.156 | 0.95f–h | 0.060 | 1085.0fg | 47.264 |
| Site 15 | 3.71k | 0.124 | 0.97e–h | 0.104 | 1008.0f–h | 7.010 |
| Site 16 | 2.21o | 0.141 | 0.64i–j | 0.003 | 808.24i | 18.236 |
| Site 17 | 1.25q | 0.113 | 0.50j | 0.070 | 930.36h | 12.838 |
| Site 18 | 2.66mn | 0.078 | 0.84h–i | 0.052 | 1030.3f–h | 122.010 |
| Site 19 | 0.72r | 0.042 | 0.85h–i | 0.003 | 1035.3f–h | 23.724 |
| Site 20 | 8.24h | 0.255 | 1.30b | 0.109 | 1346.3bc | 34.741 |
| Site 21 | 10.97f | 0.092 | 1.15b–f | 0.069 | 1336.1bc | 10.675 |
| Site 22 | 1.71p | 0.361 | 0.95f–h | 0.041 | 1002.4f–h | 113.860 |
| Minimum | 0.72 | 0.03 | 0.50 | 0.00 | 3.79 | 808.24 |
| Maximum | 17.96 | 0.53 | 1.65 | 0.30 | 122.01 | 1515.85 |
| Mean | 7.14 | 0.17 | 1.05 | 0.07 | 43.61 | 1133.70 |

Note: The mean values of the three estimated parameters in a column with the same letter are not statistically different from one another according to Duncan's multiple range test at a 0.05 significance level. SD refers to the standard deviation.

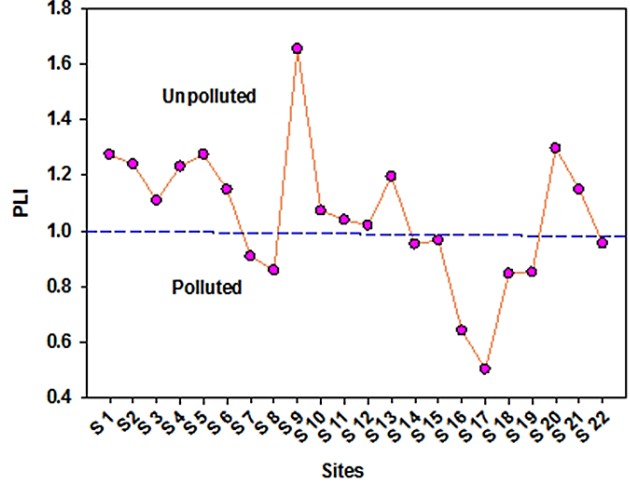

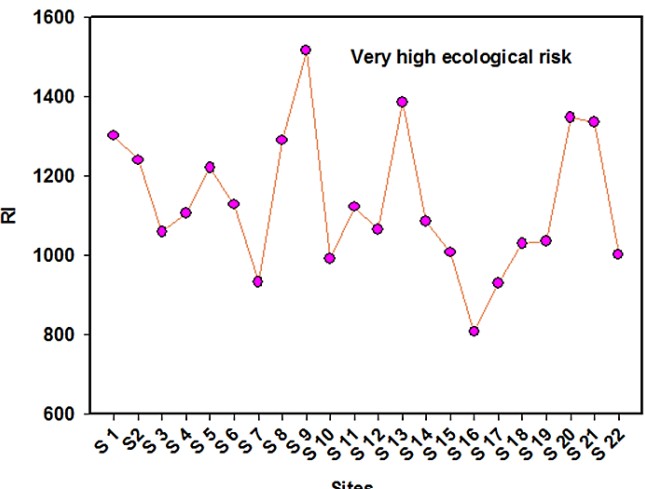

**Figure 4.** Evaluation of sediments in different sites of Lake Qaroun based on pollution load index (PLI) and potential ecological risk index (RI).

A strong relationship ($R^2$ = 0.7) between PLI and OM content of lake sediments was found (Figure 5). Also, OM content related with RI ($R^2$ = 0.5) (Figure 5). Organic matter in bottom sediment is an important food source for benthic species in aquatic habitats. Furthermore, because most organic and inorganic pollutants have a high sorption affinity for OM, it can have a substantial role in the dispersion and bioavailability of contaminants as potential harmful metals. Previously there has been evidence of trace metal enrichment in organic-rich sediments. The constant flow of residential wastewater into aquatic environments raises metal levels and organic material concentrations [15,17,101].

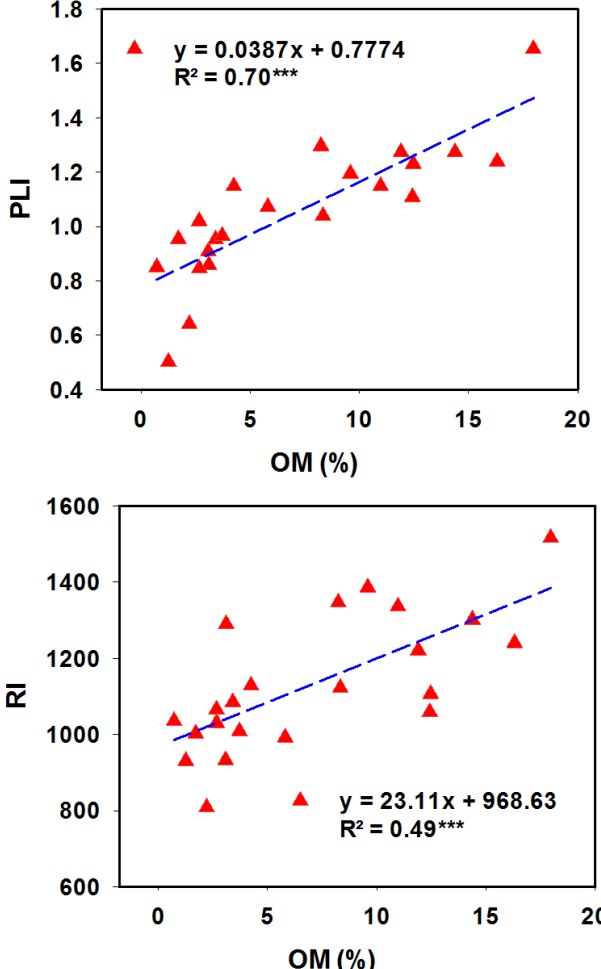

**Figure 5.** The relationships between organic matter (OM), pollution load index (PLI), and potential ecological risk index (R1) across 2 years. *** indicates high significance at $p \leq 0.001$ probability level.

### 3.3. The Variation of Spectral Measurements and Evaluation of SRIs Groups to Assess the Measured Parameters

The spectra measurements were made at various sampling stations throughout the Lake Qaroun over the 2-year investigation. Figure 6 depicts the association between wavelength and reflectance collected from different points across the lake during various dates of lake shipboard cruises. The results illustrated in the figure demonstrated significant variations in respective spectral characteristics related to the reflectance values over visible (VIS) and near infrared (NIR) ranges. The results further showed a slight difference in the combined shape of spectral signature and the wavelength-locality from 400 to 500 nm. It is also obvious that the spectral signature collected at different sampling stations varies from one location to another, showing low reflectance over the region that ranged from 400 to 500 nm (blue region), which may be a result of the absorption by the components of

OM. The variation in spectra at the green and red range (500–700 nm) showed a greater difference compared with the blue range (400–500 nm). The peak of the reflectance was recorded roughly around 800 nm (over the near infrared region; NIR). It is noticeable from the results that the difference in spectral signature between sampling stations over the NIR region is the greatest among different regions of the electromagnetic spectrum. It is also noted that there are two turning points centred around 670 and 950 nm, which would be useful for the formulation of band ratios to predict different parameters of the contents found in the collected sediment samples. The variation of the composite shape of the spectral signature obtained at different sampling points seems to be a function of absorption by OM content. Several researchers found that the spectral behaviour of sediments was dependent on physical and chemical properties, particle size distribution, and organic matter [101–104]. With increasing OM contents, the preventing role of spectral reflectivity was between 550 and 700 nm, as well as reflectance information, including 410 nm, 570 nm, 660 nm, and 870 nm, which is considered one of the most sensitive spectral bands for discriminating varying OM contents [105,106].

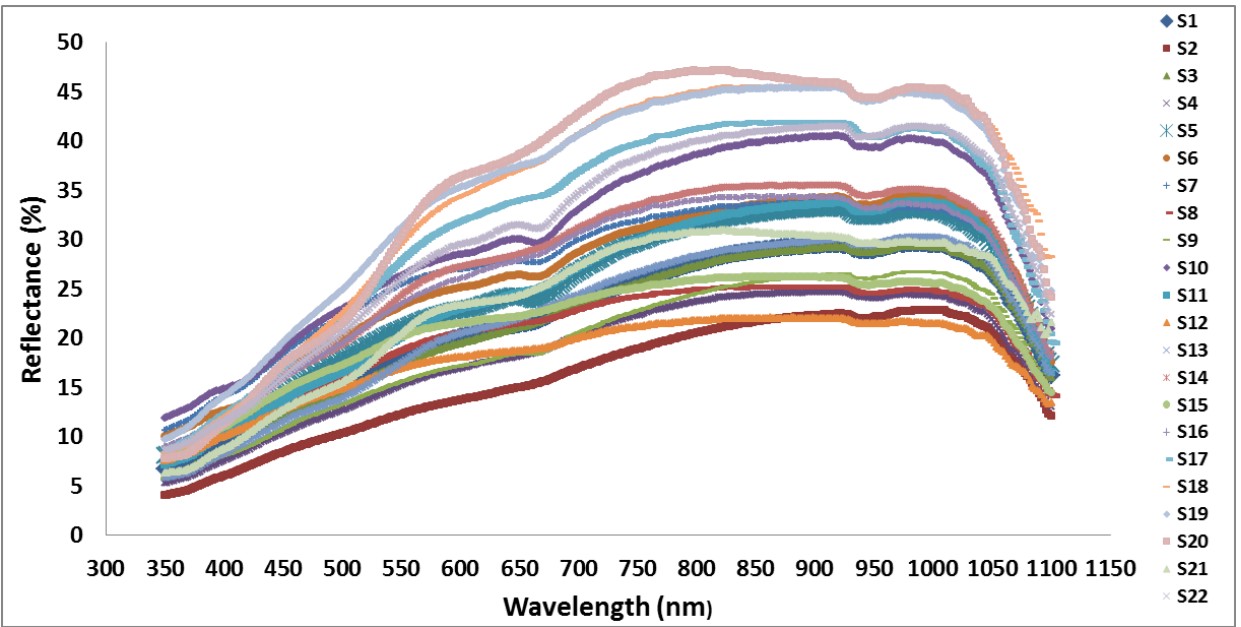

**Figure 6.** Spectra signature of the various sediment samples gathered from different locations across Lake Qaroun.

Estimating OM and pollution indices based on heavy metal levels using SRIs is a way of constructing spectral indices based on the VIS and NIR ranges, from which the model may be built. This approach may significantly improve the connection among spectral variables and metal concentrations by reducing background noise interference, supplementing information between various bands, and remarkably strengthening the association between spectral variables and various metal contents. It was possible to choose the optimal bands for estimating measured parameters, which compensated for the disadvantages of employing the entire bands. On the other hand, accurate estimate could only be performed by developing proper indices [107,108]. For that, the advantage of the present study was the optimization of two and three band SRIs by combining two or three bands, which were established by different 2-D and 3-D contour maps. In addition, there was very little information about the relationships between SRIs and PLI and RI.

Previously published, two- and three-band SRIs were evaluated in this research to assess their efficiency in the detection of OM, PLI, and RI. According to the results presented in Table 4, it is obvious that the three band spectral indices are more efficient in quantifying different investigated parameters. Generally, the previously published indices showed

poor to moderate efficiency to predict OM and poor efficiency to predict PLI and RI, since the least amount of $R^2$ for the relationship between the three parameters and PSI were found with them with a minimum $R^2$ of 0.20, 0.15, and 0.03 for predicting OM, PLI, and RI, respectively. According to the results, DVI produced the weakest relationships to predict OM, PLI, and RI. Among all tested indices, DVI index seems to be the least for predicting the three investigated parameters. Two tested band indices demonstrated moderate to strong $R^2$ for the detection of OM, PLI, and RI. The $NDSI_{682,666,692}$, $NDSI_{682,690,666}$, and $NDSI_{680,688,672}$ seem to be the optimum indices for predicting OM, PLI, and RI, respectively, with respective $R^2$ of 0.85, 0.63, and 0.58. From the results, it is obvious that the three band indices based on the wavelengths ranged from 660 to 690 are sensitive to the three tested indicators, having the greatest $R^2$ values when compared with the values produced from the newly constructed two- and three-band indices. In agreement with this study, Line et al. [33] found that RVI could be used to assess the total organic matter of the soil with $R^2 = 0.69$.

**Table 4.** Coefficients of determination ($R^2$) for the association between various SRIs and organic matter (OM), potential ecological risk index (R1), and pollution load index (PLI) across the 2-year investigation.

| SRIs | OM | PLI | RI |
|---|---|---|---|
| $\rho_{Blue}$ | 0.58 *** | 0.32 ** | 0.31 ** |
| $\rho_{Red}$ | 0.36 ** | 0.22 * | 0.13 |
| $\rho_{NIR}$ | 0.27 * | 0.15 | 0.10 |
| $NDVI_{780-550}$ | 0.38 ** | 0.18 | 0.17 |
| R-NDVI | 0.63 *** | 0.41 *** | 0.16 |
| RVI | 0.64 *** | 0.42 *** | 0.17 |
| DVI | 0.20 * | 0.17 | 0.03 |
| OSAVI | 0.36 ** | 0.21 * | 0.13 |
| $RSI_{674,666}$ | 0.74 *** | 0.52 *** | 0.34 ** |
| $RSI_{660,680}$ | 0.74 *** | 0.44 *** | 0.32 ** |
| $RSI_{676,664}$ | 0.76 *** | 0.51 *** | 0.34 ** |
| $RSI_{800,650}$ | 0.70 *** | 0.41 *** | 0.19 * |
| $RSI_{750,740}$ | 0.71 *** | 0.40 *** | 0.19 * |
| $RSI_{680,664}$ | 0.73 *** | 0.45 *** | 0.20 * |
| $RSI_{1080,650}$ | 0.59 *** | 0.48 *** | 0.21 * |
| $RSI_{822,642}$ | 0.70 *** | 0.55 *** | 0.25 * |
| $RSI_{1078,656}$ | 0.71 *** | 0.57 *** | 0.26 * |
| $RSI_{666,670}$ | 0.53 *** | 0.31 ** | 0.39 ** |
| $RSI_{900,1042}$ | 0.38 ** | 0.43 *** | 0.43 *** |
| $RSI_{912,1028}$ | 0.45 *** | 0.51 *** | 0.46 *** |
| $NDSI_{682,666,692}$ | 0.85 *** | 0.62 *** | 0.41 *** |
| $NDSI_{682,692,666}$ | 0.85 *** | 0.62 *** | 0.41 *** |
| $NDSI_{684,666,692}$ | 0.85 *** | 0.54 *** | 0.38 ** |
| $NDSI_{684,692,662}$ | 0.85 *** | 0.54 *** | 0.38 ** |
| $NDSI_{656,664,676}$ | 0.84 *** | 0.54 *** | 0.39 ** |
| $NDSI_{682,692,664}$ | 0.83 *** | 0.54 *** | 0.45 *** |
| $NDSI_{682,690,666}$ | 0.77 *** | 0.63 *** | 0.37 ** |
| $NDSI_{682,666,690}$ | 0.77 *** | 0.63 *** | 0.37 ** |
| $NDSI_{682,668,692}$ | 0.79 *** | 0.61 *** | 0.34 ** |
| $NDSI_{680,688,672}$ | 0.51 *** | 0.55 *** | 0.57 *** |
| $NDSI_{680,666,688}$ | 0.70 *** | 0.52 *** | 0.50 *** |
| $NDSI_{658,650,670}$ | 0.65 *** | 0.49 *** | 0.51 *** |

Note: *, **, and *** indicate significance at $p \leq 0.05$, $p \leq 0.01$, and $p \leq 0.001$ probability levels, respectively.

*3.4. Evaluation of ANN, PLSR and MLR, Models for Assessing Organic Matter (OM) Content, Potential Ecological Risk Index (R1) and Pollution Load Index (PLI)*

As demonstrated in Table 5, the spectral reflectance indices (SRIs) were the optimum integration for screening the uppermost variables. They showed a high ranking for estimating the tested parameters of the sediment. As presented in Table 5, the neural network was

trained with 2D and 3D indices (independent variables) for the estimate of OM, RI, and PLI (dependent variable). After that, a comparison was made between the predicted values and the reserved values, which were not done for the neural network. Our research evaluated and compared the results of several multivariate methods, concluding that multivariate methods considerably improved the predictability process. Because validation data are not comprised in the model construction process, performing the validation step independently can be regarded the most efficient technique for evaluating the regression model efficiency. The ANN- SRIs -9 was found to be the most efficient predictive model as noticed by its great performance, and there appeared to be a significant association between OM content and the superlative features of the model. This model involves nine features that had great significance for estimating OM. The model outcomes showed high efficiency with $R^2$ values of 0.999 and 0.960 for training and validation datasets, respectively. The ANN-SRIs -14 model was ranked the first for its high performance to determine PLI with high $R^2$ values of 0.999 and 0.897 in the training and validation sets, respectively. Regarding the prediction of RI, the results demonstrated that the ANN- SRIs -17 showed the most efficient model with respective $R^2$ values of 0.999 and 0.853 for training and validation sets. According to Elsherbiny et al. [109], the predicted performance was elevated to upgrade the predictability of the regression algorithms; some steps were fundamentally needed over the process of training, among which are filtering advanced-level features and also optimizing hyper-parameters of the model. As depicted in Figure 7, the neural network architectures collected senior VIs features. The most efficient neural network topology with variants can be observed in the figure. Every network architecture contains basic information including the number of unseen neuron layers, synaptic weights trained, convergence techniques, and overall errors. The network topology is mainly established with a number of hidden neuron layers and a specific interaction of input variables. The model ANN- SRIs -9, as an example, had hidden neuron layers (6,22), ANN- SRIs -14 was needed (2,14), and ANN-SRIs -17 was chosen (12,22). Advanced models of ANN- SRIs -9 are depicted in Figure 7, ANN- SRIs -14, and ANN- SRIs -17. The training process obligated 27, 5, and 274 steps, respectively for obtaining less error function. The training had a total error of 13.507, 0.029, and 4500.334, respectively. As pointed out by Thawornwong and Enke [110], to have better performance and avoid over-fitting, back-propagation algorithm with early stopping was used for training the network.

**Table 5.** Calibration and validation model outcomes of ANNs for the association between SRIs and organic matter (OM), pollution load index (PLI), and potential ecological risk index (R1) across a 2-year investigation.

| Variable | Parameters | Best Indices | Calibration | | Validation | |
|---|---|---|---|---|---|---|
| | | | $R^2$ | RMSE | $R^2$ | RMSE |
| OM | (6, 22) & Tanh | $R_{676}/R_{664}$, $NDSI_{682,666,690}$, $NDVI_{780,550}$, $NDSI_{680,666,688}$, $R_{750}/R_{740}$, RED, $NDSI_{682,668,692}$, $NDSI_{656,664,676}$, $R_{660}/R_{680}$ | 0.999 *** | 0.0003 | 0.960 *** | 0.819 |
| PLI | (2, 14) & Tanh | OSAVI, $NDSI_{682,666,692}$, $R_{800}/R_{650}$, $NDSI_{680,666,688}$, $R_{900}/R_{1042}$, $R_{1080}/R_{650}$, $R_{684,692,662}$, DVI, $R_{660}/R_{680}$, $R_{680}/R_{664}$, NDVI, $R_{750}/R_{740}$, $R_{912}/R_{1028}$, $R_{1078}/R_{656}$ | 0.999 *** | 0.002 | 0.897 *** | 0.065 |
| RI | (12, 22) & relu | $NDSI_{680,688,672}$, $R_{900}/R_{1042}$, $R_{680}/R_{664}$, $NDVI_{780,550}$, $NDSI_{658,650,670}$, DVI, $NDSI_{656,664,676}$, $R_{912}/R_{1028}$, $R_{750}/R_{740}$, $NDSI_{682,692,666}$, $NDSI_{682,666,692}$, $R_{666}/R_{670}$, $NDSI_{682,690,666}$, $NDSI_{682,666,690}$, $R_{660}/R_{680}$, $NDSI_{682,692,664}$, $NDSI_{682,668,692}$ | 0.999 *** | 1.151 | 0.853 *** | 52.70 |

Note: *** refers to significant values at $p \leq 0.001$ probability levels.

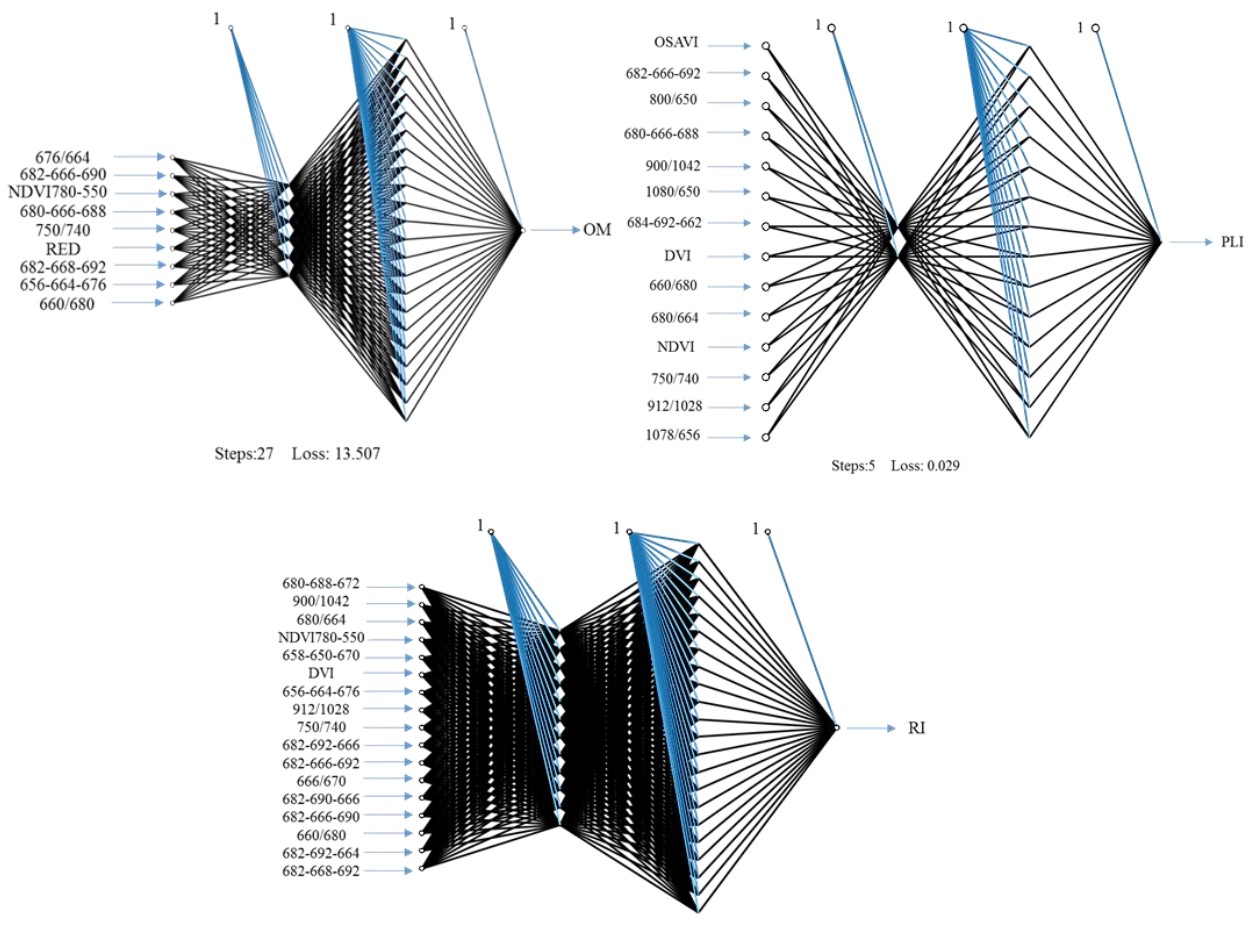

**Figure 7.** Neural network diagrams established for detecting organic matter (OM), pollution load index (PLI), and potential ecological risk index (R1).

The results presented in Tables 5–7 show the efficiency of the three tested models to predict OM, PLI, and RI, revealing that the ANN is the best model to predict different parameters (OM, PLI, and RI) with a very high determination coefficient ($R^2$ = 0.999) associated with low RMSE. Broadly, the validation datasets always produced higher determination coefficient for the three tested models. For instance, the determination coefficient values of the ANN model of calibration datasets for predicting OM, PLI, and RI were 0.999, 0.999, and 0.999, respectively, while they were 0.960, 0.897, and 0.853, respectively, for the validation dataset. As seen in the tables, PLSR produced the lowest performance especially with the calibration dataset with $R^2$ values of 0.92, 0.875, and 0.552, while with the validation dataset the PLSR produced $R^2$ values higher than with MLR for predicting PLI and RI. Figures 8–10 showed the comparison between measuring, calibrating, and validating datasets for the three investigated indicators (OM, PLI, and RI), using ANN, PLSR, and MLR. It is obvious from the three figures that there was neither obvious overfitting nor underfitting for the relationships. Moreover, the results demonstrated significant positive relationships between the quantified and the detected values with high correlations.

**Table 6.** Calibration and validation models result of PLSR for the association between SRIs and organic matter (OM), pollution load index (PLI), and potential ecological risk index (R1) across a 2-year investigation.

| Variable | PCs | Best Indices | Calibration | | Validation | |
|---|---|---|---|---|---|---|
| | | | $R^2$ | RMSE | $R^2$ | RMSE |
| OM | 3 | $R_{676}/R_{664}$, $NDSI_{682,666,690}$, $NDVI_{780,550}$, $NDSI_{680,666,688}$, $R_{750}/R_{740}$, RED, $NDSI_{682,668,692}$, $NDSI_{656,664,676}$, $R_{660}/R_{680}$ | 0.920 *** | 1.518 | 0.881 *** | 1.858 |
| PLI | 7 | OSAVI, $NDSI_{682,666,692}$, $R_{800}/R_{650}$, $NDSI_{680,666,688}$, $R_{900}/R_{1042}$, $R_{1080}/R_{650}$, $R_{684,692,662}$, DVI, $R_{660}/R_{680}$, $R_{680}/R_{664}$, NDVI, $R_{750}/R_{740}$, $R_{912}/R_{1028}$, $R_{1078}/R_{656}$ | 0.875 *** | 0.085 | 0.589 *** | 0.192 |
| RI | 3 | $NDSI_{680,688,672}$, $R_{900}/R_{1042}$, $R_{680}/R_{664}$, $NDVI_{780,550}$, $NDSI_{658,650,670}$, DVI, $NDSI_{656,664,676}$, $R_{912}/R_{1028}$, $R_{750}/R_{740}$, $NDSI_{682,692,666}$, $NDSI_{682,666,692}$, $R_{666}/R_{670}$, $NDSI_{682,690,666}$, $NDSI_{682,666,690}$, $R_{660}/R_{680}$, $NDSI_{682,692,664}$, $NDSI_{682,668,692}$ | 0.552 *** | 114.272 | 0.377 ** | 141.941 |

Note: ** and *** indicate significant values at $p \leq 0.01$ and $p \leq 0.001$ probability levels, respectively.

**Table 7.** Calibration and validation models outputs of MLR for the association between spectral reflectance indices and organic matter (OM), pollution load index (PLI), and potential ecological risk index (R1) across a 2-year study.

| Variable | Best Indices | Calibration | | Validation | |
|---|---|---|---|---|---|
| | | $R^2$ | RMSE | $R^2$ | RMSE |
| OM | $R_{676}/R_{664}$, $NDSI_{682,666,690}$, $NDVI_{780,550}$, $NDSI_{680,666,688}$, $R_{750}/R_{740}$, RED, $NDSI_{682,668,692}$, $NDSI_{656,664,676}$, $R_{660}/R_{680}$ | 0.945 *** | 1.762 | 0.824 *** | 2.38 |
| PLI | OSAVI, $NDSI_{682,666,692}$, $R_{800}/R_{650}$, $NDSI_{680,666,688}$, $R_{900}/R_{1042}$, $R_{1080}/R_{650}$, $R_{684,692,662}$, DVI, $R_{660}/R_{680}$, $R_{680}/R_{664}$, NDVI, $R_{750}/R_{740}$, $R_{912}/R_{1028}$, $R_{1078}/R_{656}$ | 0.933 *** | 0.111 | 0.496 *** | 0.208 |
| RI | $NDSI_{680,688,672}$, $R_{900}/R_{1042}$, $R_{680}/R_{664}$, $NDVI_{780,550}$, $NDSI_{658,650,670}$, DVI, $NDSI_{656,664,676}$, $R_{912}/R_{1028}$, $R_{750}/R_{740}$, $NDSI_{682,692,666}$, $NDSI_{682,666,692}$, $R_{666}/R_{670}$, $NDSI_{682,690,666}$, $NDSI_{682,666,690}$, $R_{660}/R_{680}$, $NDSI_{682,692,664}$, $NDSI_{682,668,692}$ | 0.946 *** | 70.373 | 0.604 *** | 123.448 |

Note: *** indicates significant values at $p \leq 0.001$ probability levels.

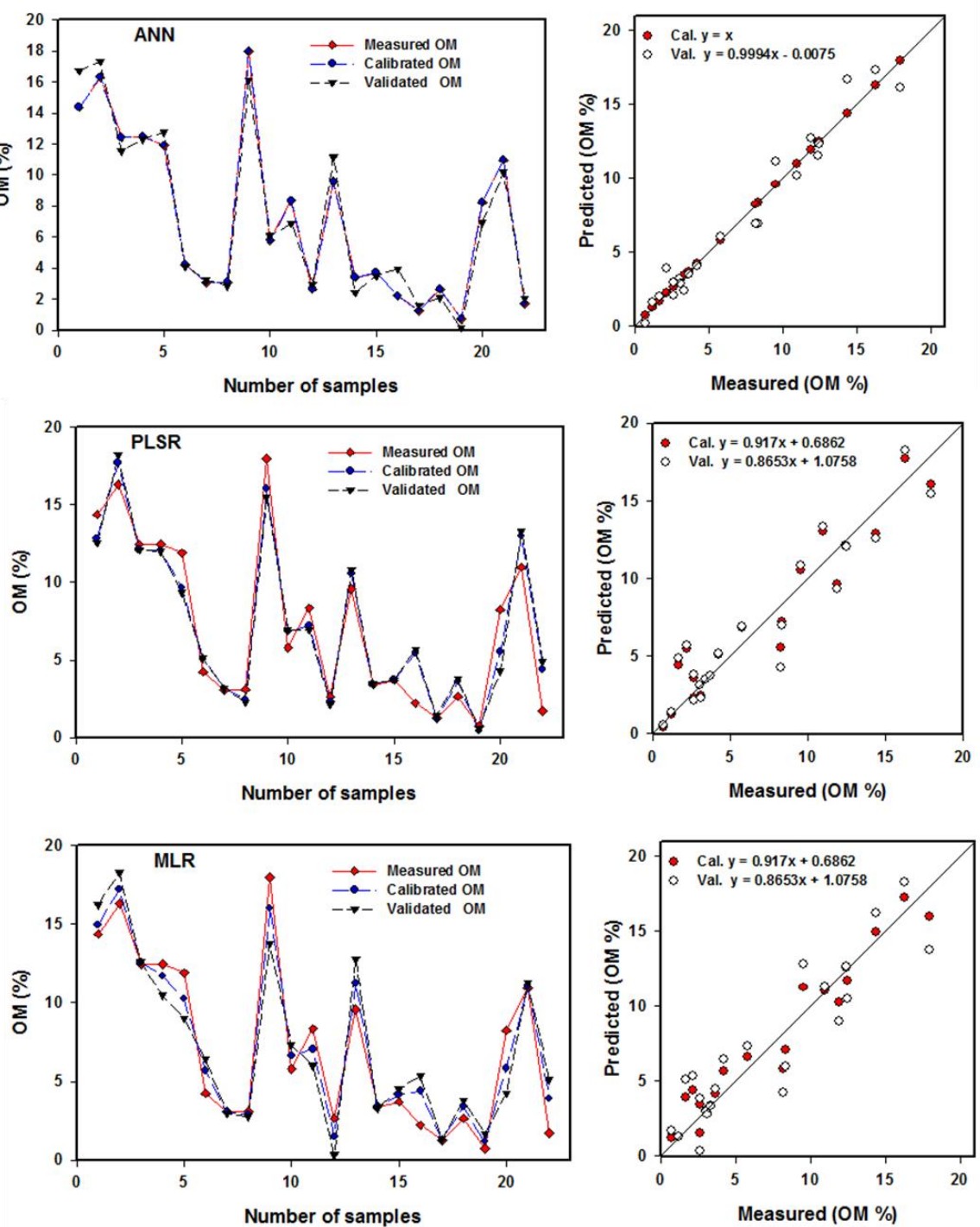

**Figure 8.** Comparing between measured, calibrated, and validated datasets for organic matter (OM) using the MLR, PLSR, and ANN models based on several SRIS.

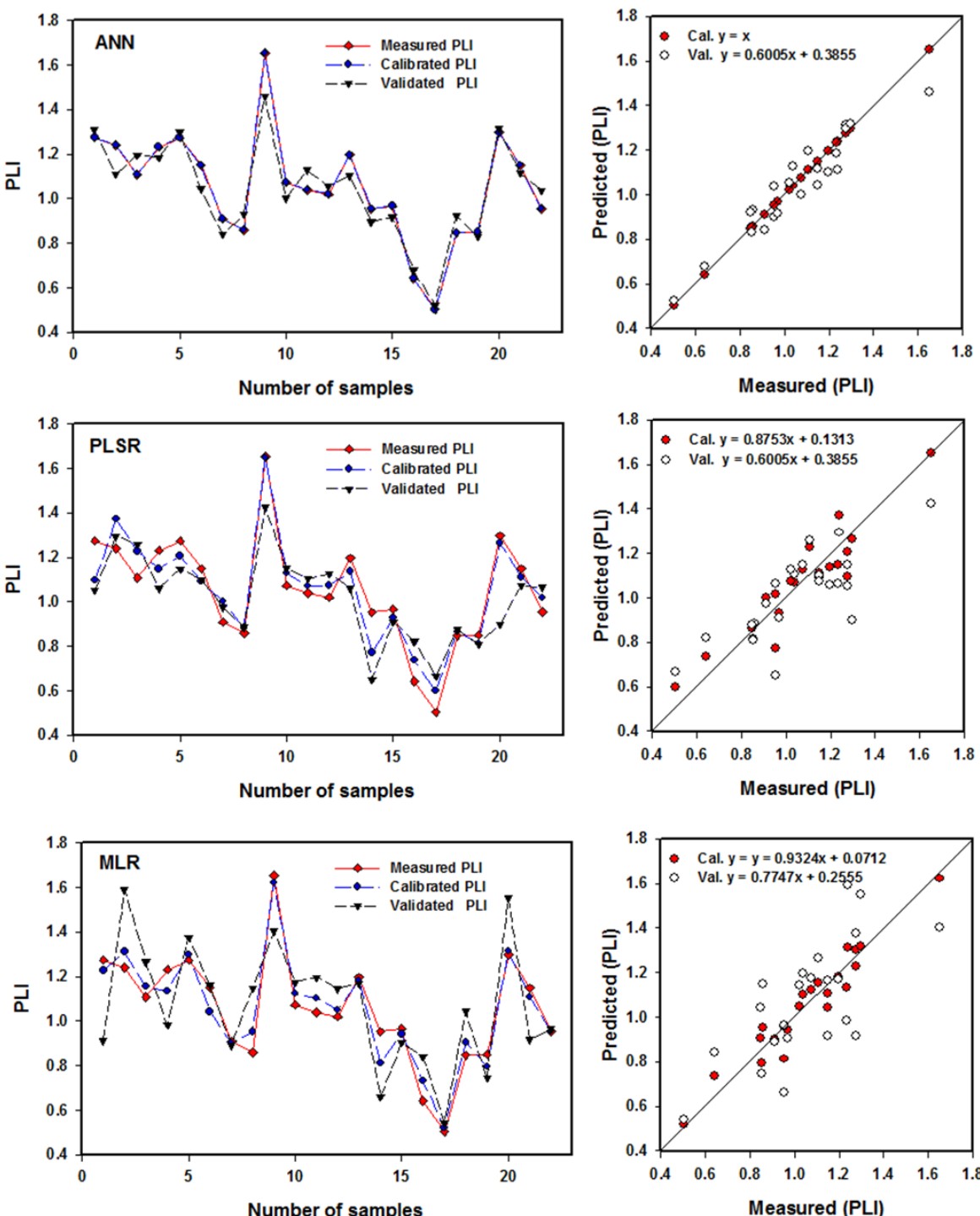

**Figure 9.** Results of the comparison between measured, calibrated, and validated datasets for pollution load index (PLI) using the ANN, PLSR, and MLR models based on several SRIS.

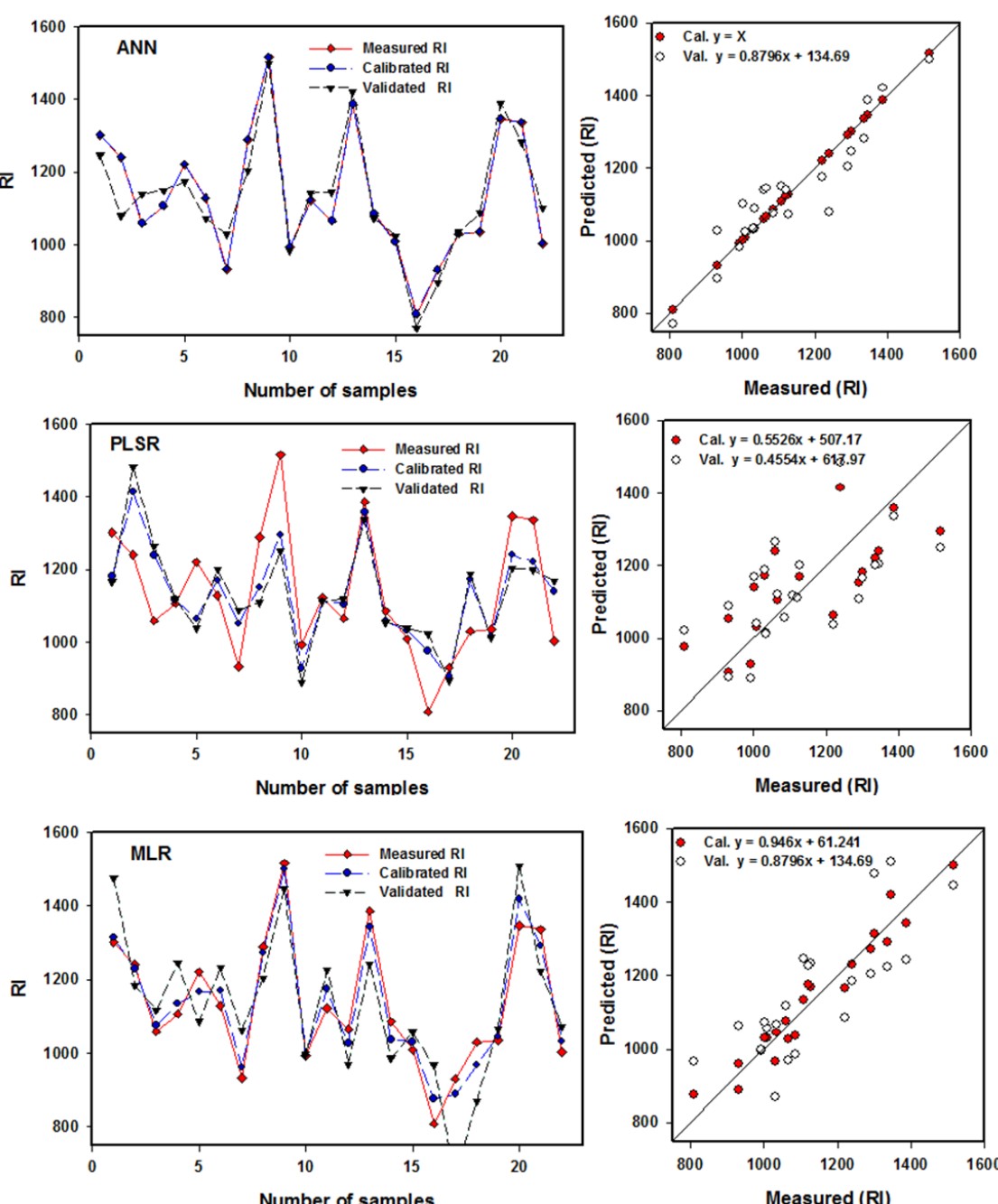

**Figure 10.** Association between measured, calibrated, and validated datasets for potential ecological risk index (R1) using the MLR, PLSR, and ANN models based on several SRIS.

## 4. Conclusions

Multi-contamination indices, spectral reflectance indices, and data driven and multivariate modelling, were used as inexpensive approaches for assessing PTMs in bottom sediments. As a result, we anticipated their ecological dangers and assisted the decision-makers with environmental protection and remedial procedures. The organic-rich layers of the lake's eastern and central regions have the greatest levels of all the metals studied. The lowest concentrations were often found in western and north-west sediment samples characterised by a lower percentage of organic matter. The southern and eastern sides of the lake's central section were polluted (PLI > 1), but the western and west-northern areas remained clean (PLI < 1). RI's findings were that all the examined sediments pose a very

high ecological risk (RI > 600). It is obvious that the three-band spectral indices are more efficient in quantifying different investigated parameters. In addition, the results showed that the efficiency of the three tested models (ANN, PLSR, and MLR) to predict OM, PLI, and RI, revealed that the ANN is the best model to predict different parameters (with a very high determination coefficient ($R^2$ = 0.999) associated with low RMSE.

Our future work will hopefully involve a light hand-held apparatus based on two and three band ratios that may be used for the detection of bottom sediment quality without disturbing the lake's bottom. Extrapolating the results of in situ spectroscopy surveys using advanced newly launched satellites would be a promising tool in lake monitoring systems in our future work. The newly launched satellites with high spectral capabilities (high and medium resolution) can successfully detect sediment characteristics effectively at a large scale. The findings of this research study would be adequate to provide a potential reference for estimating OM and two pollution indices; also, this research offers technological assistance for long-term monitoring and assessment of the sediment quality of lakes.

**Supplementary Materials:** The following are available online at https://www.mdpi.com/article/10.3390/w14060890/s1, Table S1. Average of metal concentrations (ppm) and total organic matter (OM) in bottom sediment of Lake Qaroun in first year; Table S2. Average of metal concentrations (ppm) and total organic matter (OM) in bottom sediment of Lake Qaroun in second year.

**Author Contributions:** Conceptualization, A.H.S. and S.E.; methodology, M.M.A.E.-S., A.H.S. and S.E.; software, S.E., A.H.S., O.E., H.H. and M.M.A.E.-S.; validation, A.H.S., O.E., S.E., A.H.E. and M.G.; formal analysis, A.H.S., O.E., S.E. and M.M.A.E.-S.; investigation A.H.S., S.E., A.H.E. and M.M.A.E.-S.; resources, F.S.M., E.M.E.; data duration, A.H.S., M.G., S.E., F.S.M. and M.M.A.E.-S.; writing—original draft preparation, A.H.S., S.E., O.E., A.H.E. and M.M.A.E.-S.; writing—review and editing, A.H.S., S.E., A.H.E., O.E., A.S.Q., E.M.E., A.S.E.-K., M.A.T. and M.M.A.E.-S.; supervision, E.M.E.; project administration E.M.E.; funding acquisition, M.A.T. All authors have read and agreed to the published version of the manuscript.

**Funding:** The Deanship of Scientific Research at King Khalid University, supported this study with grant number RGP. 1/182/43.

**Institutional Review Board Statement:** Not applicable.

**Informed Consent Statement:** Not applicable.

**Data Availability Statement:** Data are contained within the article.

**Conflicts of Interest:** The authors declare no conflict of interest.

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
