# Peer review of "Utilization of Pollution Indices, Hyperspectral Reflectance Indices, and Data-Driven Multivariate Modelling to Assess the Bottom Sediment Quality of Lake Qaroun, Egypt"

_water, doi:10.3390/w14060890_

Round 1
Reviewer 1 Report
This study demonstrated the application of PLI and RI in evaluating the ecological risk of PTMs in the bottom sediment of Qaroun Lake and optimized the two and three-band spectral indices and assessed their accuracy.
The paper is very well written and structured. The introduction is very informative, materials and methods are well described. An appropriate statistical analysis has been conducted, which makes the results reliable. The conclusion is also well supported by the achieved results.
I only have two comments as below:
1. I think that the abstract is too long. It is very informative, but still too wordy. I suggest making it more summarized. It is not necessary to explain everything in the abstract. Please just highlight the most important aspects. Keeping it below 300 words is an art for the authors.
2. Line 217: I invite the authors to discuss the time of processing sampling. Have you done the sampling on autumns only? Why? Does it have a specific reason? Please introduce the readers to the specific climate of the lake and explain why autumn is the best season for conducting sampling.
Author Response
Dear Prof. Dr. Jean-Luc PROBST,
Editor-in-Chief - Water
Please find attached the revised manuscript titled: Utilization of Pollution indices, Hyperspectral Reflectance in-dices, Data-driven, and Multivariate Modelling to Assess the Bottom Sediment Quality of Lake Qaroun, Egypt; water-1626443, authored by Ali H. Saleh, Salah Elsayed, Mohamed Gad, Adel H. Elmetwalli, Osama Elsherbiny, Hend Hussein, Farahat S. Moghanm, Amjad S. Qazaq, Ebrahem M. Eid, Aziza S. El-Kholy, Mostafa A. Taher and Magda M. Abou El-Safa.
On behalf of my co-authors, I thank you very much for giving us the opportunity to revise our manuscript. We have carefully studied the reviewers’ comments and have made revisions that are tracked in the revised version of the manuscript. We have tried our best to revise our manuscript according to the reviewers’ comments. Please find attached the revised version of our manuscript, which we would like to submit for your kind consideration. Once again, we would like to express our great appreciation to you and the reviewers for the comments on our manuscript. Please find below our detailed responses to each of the points raised.
---------------------------------------------------------------------------------------------------------------------
Comments of Reviewer# 1:
This study demonstrated the application of PLI and RI in evaluating the ecological risk of PTMs in the bottom sediment of Qaroun Lake and optimized the two and three-band spectral indices and assessed their accuracy. The paper is very well written and structured. The introduction is very informative, materials and methods are well described. An appropriate statistical analysis has been conducted, which makes the results reliable. The conclusion is also well supported by the achieved results.
Response: Thanks so much Sir for your time and for constructive comments and suggestions. We hope that we could address your questions/comments by the explanations and revisions made in the manuscript.
---------------------------------------------------------------------------------------------------------------------
I only have two comments as below:
- I think that the abstract is too long. It is very informative, but still too wordy. I suggest making it more summarized. It is not necessary to explain everything in the abstract. Please just highlight the most important aspects. Keeping it below 300 words is an art for the authors.
Response: Many thanks Sir for comment. The abstract was summarized as possible by removed unnecessary sentences.
---------------------------------------------------------------------------------------------------------------------
- Line 217: I invite the authors to discuss the time of processing sampling. Have you done the sampling on autumns only? Why? Does it have a specific reason? Please introduce the readers to the specific climate of the lake and explain why autumn is the best season for conducting sampling.
Response: Yes, the samples were collected in autumns through two years to avoid the extreme weather in summer and winter. The specific climate of the lake was added from line 224 to line 228 under section (2.1. Study Area).
---------------------------------------------------------------------------------------------------------------------
I would appreciate if the revised version of our manuscript would be considered for publication in Water.
Sincerely,
Ebrahem M. Eid
Kafrelsheikh University, Kafr El-Sheikh, Egypt

Reviewer 2 Report
The topic Utilization of Pollution indices, Hyperspectral Reflectance indices, Data-driven, and Multivariate Modelling to Assess the Bottom Sediment Quality of Lake Qaroun, Egyptis potentially interesting, however, there are some issues that should be addressed by the authors: The Introduction" sections can be made much more impressive by highlighting your contributions. The contribution of the study should be explained simply and clearly. The authors should further enlarge the Introduction with current work about artificial intelligence techniques to improve the research background, for example: Experimental setup for online fault diagnosis of induction machines via promising IoT and machine learning: Development of an IoT Architecture Based on a Deep Neural Network against Cyber Attacks for Automated Guided Vehicles; Effective IoT-based Deep Learning Platform for Online Fault Diagnosis of Power Transformers Against Cyberattack and Data Uncertainties.
Clarify how you handle the constraints of the system
Clarify the practical implementation of the proposed strategy according to the cost
Clarify how you adjust the parameters of the proposed approach
Increase the resolution of Figure 7
Conclusion section should be rearranged. According to the topic of the paper, the authors may propose some interesting problems as future work in the conclusion.
This study may be proposed for publication if it is addressed in the specified problems.
Author Response
Dear Prof. Dr. Jean-Luc PROBST,
Editor-in-Chief - Water
Please find attached the revised manuscript titled: Utilization of Pollution indices, Hyperspectral Reflectance in-dices, Data-driven, and Multivariate Modelling to Assess the Bottom Sediment Quality of Lake Qaroun, Egypt; water-1626443, authored by Ali H. Saleh, Salah Elsayed, Mohamed Gad, Adel H. Elmetwalli, Osama Elsherbiny, Hend Hussein, Farahat S. Moghanm, Amjad S. Qazaq, Ebrahem M. Eid, Aziza S. El-Kholy, Mostafa A. Taher and Magda M. Abou El-Safa.
On behalf of my co-authors, I thank you very much for giving us the opportunity to revise our manuscript. We have carefully studied the reviewers’ comments and have made revisions that are tracked in the revised version of the manuscript. We have tried our best to revise our manuscript according to the reviewers’ comments. Please find attached the revised version of our manuscript, which we would like to submit for your kind consideration. Once again, we would like to express our great appreciation to you and the reviewers for the comments on our manuscript. Please find below our detailed responses to each of the points raised.
---------------------------------------------------------------------------------------------------------------------
Comments of Reviewer# 2:
The topic Utilization of Pollution indices, Hyperspectral Reflectance indices, Data-driven, and Multivariate Modelling to Assess the Bottom Sediment Quality of Lake Qaroun, Egyptis potentially interesting, however, there are some issues that should be addressed by the authors: The Introduction" sections can be made much more impressive by highlighting your contributions. The contribution of the study should be explained simply and clearly.
Response: We greatly appreciate your critical observations as well as your constructive and helpful comments. We hope that we could address your questions/comments by the explanations and revisions made in the manuscript. The contribution of this study was highlighted in the introduction of the manuscript from lines 163 to 167 as well as from lines 197 to 202. As stated in the introduction, we focused on the study's novelty because relatively little attention has been given to employing spectral band ratios, ANN, PLSR, and MLR models to detect the quality of Lake Qaroun's bottom sediment. Monitoring the quality bottom sediments via remote sensing would be a robust tool that is quicker than point-sampling procedure. The introduction covered the objectives of this study. Firstly, we presented environmental hazard of potential toxic element in lake and explained the traditional methods of environmental monitoring from lines 65 –125. Secondly, we presented the spectral reflectance indices of remote sensing data, and we highlighted the novelty of these tools in this study from line 128 to line 162. Also, we presented the three ANN, PLSR, and MLR models and we highlighted the novelty of these models in this study.
---------------------------------------------------------------------------------------------------------------------
The authors should further enlarge the Introduction with current work about artificial intelligence techniques to improve the research background, for example: Experimental setup for online fault diagnosis of induction machines via promising IoT and machine learning: Development of an IoT Architecture Based on a Deep Neural Network against Cyber Attacks for Automated Guided Vehicles; Effective IoT-based Deep Learning Platform for Online Fault Diagnosis of Power Transformers Against Cyberattack and Data Uncertainties.
Response: In general, according to your suggestion we wrote in brief about these technologies in the introduction because they are not our target in this study. As you may be aware IoT Architecure based on deep Neural Network require a large number of observations and the use of smart systems that are not readily accessible to us.
---------------------------------------------------------------------------------------------------------------------
Clarify how you handle the constraints of the system
Response: Many thanks Sir for comment. If you mean the spectral data acquisition system, yes there are some constraints. One of the most constraints is the changing weather over the time. Although, we planned to conduct the measurements on cloud-free days, sometimes you cannot control it. To avoid misleading data, we tried to measure the spectral reflectance in around midday. This was explained from line 298 to line 300 under section (2.4. In Situ Ground-Based Reflectance Measurements).
---------------------------------------------------------------------------------------------------------------------
Clarify the practical implementation of the proposed strategy according to the cost
Response: In terms of cost, the spectra collection can be a cost-effective approach for monitoring bottom sediment quality since the traditional procedures mainly based on collecting samples and laboratory work which despite accurate, it takes very long time (time consuming) and costly when covering a large area. This was explained in abstract from line 33 to line 36 and in introduction section from line 128 to 130.
---------------------------------------------------------------------------------------------------------------------
Clarify how you adjust the parameters of the proposed approach
Response: The optimal hyperparameters for the ANN model were chosen after extensive testing. The number of neurons in two hidden layers, as well as the activation function, were the ANN parameters. In general, the spectral indices were fed to the model at random in the first loop, the low-level features were deleted during each loop, and the excellent features were sorted in an ascending order based on the largest contribution to the model over the subsequent loops. During the looping process, only the best hyperparameters were selected and the remainder were discarded from consideration. The output of the ANN was then compared in order to determine high-ranking variations and superior hyperparameters with the lowest RMSEV that might enhance the prediction.
---------------------------------------------------------------------------------------------------------------------
Increase the resolution of Figure 7
Response: Thanks for your recommendation; we redrew the figure again to have higher resolution.
---------------------------------------------------------------------------------------------------------------------
Conclusion section should be rearranged. According to the topic of the paper, the authors may propose some interesting problems as future work in the conclusion.
Response: Our future work proposes making a light hand-held apparatus based on two and three band ratios that may be used for the detection of bottom sediment quality without disturbing the lake's bottom. Extrapolating the results of in situ spectroscopy surveys using advanced newly launched satellites would be promising tool in lake monitoring systems in our future work. The newly launched satellites with high spectral capabilities (high and medium resolution), can successfully detect sediment characteristics effectively at a large scale. This was written from line 626 to line 632.
---------------------------------------------------------------------------------------------------------------------
I would appreciate if the revised version of our manuscript would be considered for publication in Water.
Sincerely,
Ebrahem M. Eid
Kafrelsheikh University, Kafr El-Sheikh, Egypt

Round 2
Reviewer 2 Report
The authors handled all comments. Thank you